



# Unexpected high frequency of nocturnal surface ozone enhancement events over China: Characteristics and mechanisms

Cheng He[1, 2], Xiao Lu[1, 2*], Haolin Wang[1, 2], Haichao Wang[1, 2], Yan Li[3], Guowen He[1, 2], Yuanping He[1, 2], Yurun Wang[4], Youlang Zhang[1, 2], Yiming Liu[1, 2], Qi Fan[1, 2], Shaojia Fan[1, 2*]

[1]School of Atmospheric Sciences, Sun Yat-sen University, and Key Laboratory of Tropical Atmosphere-Ocean System, Ministry of Education, Zhuhai, China
[2]Guangdong Provincial Observation and Research Station for Climate Environment and Air Quality Change in the Pearl River Estuary, Southern Marine Science and Engineering Guangdong Laboratory (Zhuhai), Zhuhai, China
[3]School of Environmental Science and Engineering, China West Normal University, Nanchong, China
[4]Department of Land, Air, and Water Resources, University of California Davis, Davis, USA

*Correspondence to*: Xiao Lu (luxiao25@mail.sysu.edu.cn) and Shaojia Fan (eesfsj@mail.sysu.edu.cn)

**Abstract.** Surface ozone concentrations typically peak in daytime driven by active photochemical production and decrease gradually after sunset by chemical destruction and dry deposition. Here, we report that nocturnal ozone enhancement (NOE, defined as ozone increase by at least 5 ppbv hour$^{-1}$ in one of any two adjacent hours in 20:00-06:00 local time) events are observed at multiple monitoring sites in China at a high frequency that has not been recognized in previous studies. We present an overview of the general characteristics of NOE events in China and explore the possible mechanisms based on six-year observations from the national monitoring network. We find that the annual mean frequency of NOE events is 41±10% averaged over all 814 Chinese sites in 2014-2019, which is 46% larger than those over Europe and US. The NOE event frequency is higher in industrialized city clusters (>50%) than regions with lighter ozone pollution, and is higher in the warm (46%) than cold season (36%), consistent with the spatiotemporal evolution of ozone levels. The mean ozone peak during NOE events reaches 37±6 ppbv in the warm season. The ozone enhancements are within 5-15 ppbv hour$^{-1}$ in 85% of the NOE events, but in about 10% of the cases the ozone increases can exceed 20 ppbv hour$^{-1}$. We propose that the high photochemistry-induced ozone in the daytime provides rich ozone source in the nighttime residual layer, determining the overall high frequency of NOE events in China, and then the enhanced atmospheric mixing triggers NOE events by allowing the ozone-rich air in the residual layer to mix into the nighttime boundary layer. This is supported by our analyses that 70% (65%) of the NOE events are associated with increases in friction velocity (planetary boundary layer height), indicative of enhanced atmospheric mixing, and also supported by the observed sharp decreases in surface $NO_2$ and CO concentrations with ozone increases in NOE events, a typical signal of mixing with air in the residual layer. Three case studies in Beijing and Guangzhou show that synoptic processes such as convective storms and low-level jets can lead to the NOE event by aggravating vertical mixing. Horizontal transport of ozone-rich plumes may also be a supplementary driver of NOE event. Our results summarize for the first time the characteristics and mechanism of the NOE events in China based on nationwide and long-term observations, and call for more direct measurements and modeling studies on the nighttime ozone evolution from surface to the residual layer.



## 1 Introduction

Surface-layer ozone is a major air pollutant and controls the oxidation capacity of the atmospheric boundary layer. Ozone is
35 mainly generated from the sunlight-driven photochemical reactions of nitrogen oxides ($NO_x$, $NO \pm NO_2$), carbon monoxide
(CO), and volatile organic compounds (VOCs) that released from anthropogenic and natural sources at the surface (Sillman,
1995; Jacob, 2000; Monks et al., 2015). It is also transported downward from the upper troposphere or even stratosphere with
mesoscale dynamic processes triggering deep stratospheric incursions (Stohl et al., 2003; Škerlak et al., 2014; Wang et al.,
2020). Surface-layer ozone levels in polluted regions typically increase in daytime fostered by photochemical formation in the
40 presence of solar radiation and emissions of precursors, and decrease gradually after sunset by chemical destruction (i.e. $NO_x$
titration) and dry deposition, shaping the well-established single daytime peak pattern of ozone diurnal cycle (Lin et al., 2008;
Strode et al., 2019). Here, we report that nocturnal ozone enhancement events are observed at multiple monitoring sites in
China at a high frequency that have not been recognized in previous studies.

Analyses of the nocturnal ozone enhancement events and associated nighttime ozone peaks have important implications for
human health, vegetation growth, and nocturnal atmospheric chemistry. Ozone toxicity is capable of stimulating the lining of
the lung and other surfaces in the respiratory tract and thus increases both short-term and long-term risks of respiratory and
circulatory mortality (Turner et al., 2016; Fleming et al., 2018). With rapid urbanization and the consequent increasing numbers
of population travelling and working nighttime, exposure to enhanced nocturnal ozone levels may increase the threats of
50 cardiovascular and respiratory diseases. Surface ozone is also harmful to vegetation growth and crop yields through damaging
plant leaf cells (Yue et al., 2017; Lefohn et al., 2018; Feng et al., 2022; Li et al., 2022). Although ozone-induced damages to
vegetation are most prominent at daytime due to more vigorous metabolic activities and higher ozone concentrations, nighttime
ozone also impairs stomatal function of plants such as tobacco which is more active at night (De Moraes et al., 2001; Cordeiro
et al., 2021). Meanwhile, elevated nocturnal ozone may enhance the oxidation capacity that promotes the formation of
55 secondary pollutants and affect atmospheric chemistry of the following day (Millet et al., 2016; Caputi et al., 2019; Zhao et
al., 2020).

A number of studies have reported episodic nocturnal ozone enhancement events at individual sites around the world (Eliasson
et al., 2003; Hu et al., 2012; Hu et al., 2013; Klein et al., 2014; Kulkarni et al., 2016; Caputi et al., 2019; Zhu et al., 2020; He
et al., 2021).The reported nocturnal ozone enhancement were mostly ranging from 5 to 30 ppbv, but can be up to 70 ppbv in
an extreme event observed in summer 2013 over the North China Plain (Jia et al., 2015). The mean nocturnal ozone peaks
during these episodes may exceed 40 ppbv (Zhu et al., 2020), sufficiently high to threat human and vegetation health. Previous
studies have identified that horizontal transport of ozone-rich air from polluted regions, and enhanced turbulence and vertical
mixing which can be triggered by synoptic processes such as convective storms and low-level jets, are the mechanisms for
driving the nocturnal ozone enhancement (Reitebuch et al., 2000; Hu et al., 2013; Klein et al., 2014; Caputi et al., 2019). These


studies, however, focused on episodic cases at only one or few sites and were limited to timescales of days or months. A comprehensive view on the general characteristics and mechanisms of the nocturnal ozone enhancements based on multi-year and large-scale observation monitoring network is missing.

In this study, we examine the frequency, peak ozone values, magnitude, and trend of the nocturnal ozone enhancement (NOE) events over China, based on six-year (2014-2019) ozone observations from the nationwide monitoring network. We compare the frequency of NOE events in China with that in the United States and Europe in the same period to demonstrate the higher frequency of NOE events over China than the other two regions which has not been recognized in previous studies. We then explore the potential drivers of the NOE events over China, by examining the evolution of nighttime chemical components

and meteorological factors indicative of atmospheric mixing ability in the NOE events. Our study provides the first overview of the nocturnal ozone enhancement events in China, which adds new insights to surface ozone evolution and may have important implications for designing ozone regulation strategies.

## 2 Data and methodology

### 2.1 Surface observations of ozone and air pollutants over China, Europe, and the United States

The nationwide hourly observations of surface air pollutants (ozone, $NO_2$, and CO) in mainland China are obtained from the China National Environmental Monitoring Center (CNEMC) network (http://106.37.208.233:20035/, last access: 15 April 2022). The network was initiated from 2013 at 74 key cities for air pollution monitoring, and expanded rapidly to more than 400 cities afterwards (Figure S1). The dataset has been applied in many studies for analyzing the diurnal cycle, season shifts, and interannual variations of ozone levels over Chinese cities (Wang et al., 2017b; Guo et al., 2019; Liu and Wang, 2020; Lu

et al., 2020; Li et al., 2021). Here we use the hourly data from 2014 to 2019 at 814 sites covering 74 cities with continuous ozone observations in the six-year period (Figure S1). The measurements are reported at Beijing time (UTC +8) in unit of μg m-3, and we transfer them to mixing ratios (ppbv) at local time (LT) based on the time zone of sites. We carry out data quality controls for observations of all pollutants to remove potential data outliers following Lu et al. (2018) and Jiang et al. (2021) (Supplementary information).

We use additionally hourly surface ozone observations from the United States (US) and Europe (EU) in the same 2014-2019 period for a global comparison of the frequency of nocturnal ozone enhancement events. Ozone observations at 762 sites over US and 1880 sites over EU are obtained from the US Environmental Protection Agency (https://aqs.epa.gov/aqsweb/airdata/download_files.html, last access: 15 April 2022) and the European Environment Agency

(https://discomap.eea.europa.eu/Index/, last access: 15 April 2022), respectively. We also convert the measurements to local time in unit of ppbv and apply the same data quality control procedures to these observations as to the CNMEC observations.



## 2.2 Meteorological parameters and ozone from the ERA5 dataset

We apply three-dimensional fields of meteorological parameters including temperature, relative humidity, horizontal and vertical wind speed and direction on pressure levels, and two-dimensional fields of planetary boundary layer height, and friction velocity from the ERA5 dataset, i.e. the fifth generation of the European Centre for Medium-Range Weather Forecasts (ECMWF) atmospheric reanalysis of the global climate (https://cds.climate.copernicus.eu/#!/home, last access: 15 April 2022). The ERA5 dataset combines observations of meteorological parameters across the world with model outputs, and has been extensively evaluated with independent observations of key meteorological parameters in the troposphere (Hersbach et al., 2020). The three-dimensional ozone fields from the ERA5 dataset are also utilized for the analysis of ozone vertical transport. We use the ERA5 data with the horizontal resolution at 0.25°×0.25°, vertical resolution of 25 hPa, and temporal resolution at 1 hour. We sample the ERA5 data at the grid of a monitoring site for site-level analyses.

## 2.3 Definition of nocturnal ozone enhancement event and related magnitude and nocturnal ozone peaks

We define the period of 20:00 (LT) to 06:00 (LT) of the next day as the nighttime period. Following previous studies of Eliasson et al. (2003) and Zhu et al. (2020), we define a nocturnal ozone enhancement (NOE) event if ozone concentration at a site increases by more than 5 ppbv ($\Delta O_3/\Delta t > 5\ ppb\ hour^{-1}$) in one of any two adjacent hours in the nighttime period. We also define the corresponding magnitude of an NOE event as the maximum of ozone enhancement in the two adjacent hours in the nighttime period (maximum of $\Delta O_3/\Delta t$). The nocturnal ozone peak in an NOE event is defined as the maximum of ozone concentrations in the nighttime period. For comparison, we define a non-nocturnal ozone enhancement (NNOE) event if the maximum of ozone change in all adjacent two hours in the nighttime period is less than 1 ppbv (maximum of $\Delta O_3/\Delta t < 1\ ppb\ hour^{-1}$).

## 3 Results and discussion

### 3.1 Frequency and magnitude of NOE events from 2014 to 2019

Figure 1a compares the annual frequency of the NOE events over China, Europe, and the continental US averaged over 2014-2019. We estimate the frequency of NOE events of 41±10% (mean ± standard deviation, 814 sites) averaged for all Chinese sites in 2014-2019, with 140 sites showing a frequency of 50% or higher. The annual frequency of NOE events over China are 46% larger than those over Europe (28±12%, 1880 sites) and US (28±13%, 762 sites), and this high frequency of NOE events is largely unrecognized among current ozone studies in China. Spatially, sites with high frequency of NOE events (>50%) are widely distributed across the industrialized regions such as the North China Plain (NCP), Yangtze River Delta (YRD), Pearl River Delta (PRD), and Sichuan Basin (SCB), and also scattered at regions of western China, Mediterranean, and western US. These regions have been previously identified as hotspots of severe surface ozone pollution due to either high anthropogenic emissions of ozone precursors or natural background ozone (Zhang et al., 2014; Wang et al., 2017a; Lu et al., 2018; Lu et al.,





2021). As will be discussed later, the spatial pattern of NOE event frequencies is closely related to the afternoon (14-17 LT) ozone concentrations measured at the surface as shown in Figure 1b.

Figure 2 compares the frequency and nocturnal ozone peak of NOE events in the warm (April-September) and cold (other months) season across the Chinese monitoring sites. We see notable seasonal shifts in the spatial distributions of NOE event frequencies, from 46 ± 11% averaged for all sites in the warm season to 36 ± 10% in the cold season. Regionally, the frequencies of NOE events are higher in the NCP during the warm season, while in the PRD region the frequencies are higher in the cold season (largely driven by October), though there are also large variabilities even at adjacent sites. Sites at western

China typically show high frequencies of NOE events throughout the year. These seasonal shifts in the NOE event frequency at different regions are also consistent with the seasonal evolution of ozone concentrations. Previous studies have documented the warm-season peak of ozone over the North China driven by active photochemical formation of ozone in the presence of intensive anthropogenic emissions, and the autumn peak of ozone over the PRD due to the influences of summer monsoon (Lu et al., 2018; Gao et al., 2020). We do not find consistent trends of the NOE frequency across the cities either in the warm or

cold season in 2014-2019 (Figure S2).

We then examine the peak ozone concentrations in the NOE events in Figure 2 to quantify the influences of NOE events on nighttime ozone levels. The NOE-induced nocturnal ozone peaks are 37±6 ppbv and 31±6 ppbv averaged in the warm and cold season, respectively, significantly higher than those in the NNOE events of 17±11 ppbv and 10± 9 ppbv. High nocturnal

peak ozone concentrations of more than 50 ppbv in the NOE events are found over the NCP in the warm season. These ozone values can reach over 80% of the corresponding daytime mean ozone concentrations, and are sufficiently high compared to the threshold above which ozone exposures are expected to exert negative effect on human health and vegetation growth (Lefohn et al., 2018). The mean nighttime peak ozone concentration in the NOE events in Beijing is 40 ppbv in the warm season averaged over 2014 to 2019, comparable with Zhu et al. (2020) who found 68.1μg m$^{-3}$ (approximately 34 ppbv)

averaged in May-September from 2014 to 2015. We also find that at more than 70% sites the peak ozone concentrations in the NOE events show positive trends from 2014-2019 (Figure S3). This may also reflect the rapid increasing nighttime ozone levels in China in this period as reported in Lu et al. (2020).

Figure 3 presents the magnitude of the NOE events (i.e. the maximum of $\Delta O_3/\Delta t$ in the NOE events) over China in the warm

season, quantifying the extent of ozone increases in the two adjacent hours during NOE events. We focus on warm season here because it shows significantly higher frequency of NOE events in most regions (Figure 2). We find that about 85% of the maximum ozone enhancement during NOE events are within 5-15 ppbv hour$^{-1}$, but in about 10% of the NOE events ozone enhancement can exceed 20 ppbv hour$^{-1}$ within one hour that may significantly change the ozone diurnal cycle. The frequency of small ozone enhancement in the range of 5 to 10 ppb hour$^{-1}$ during NOE events is higher in the southern than in the northern





of China, but the pattern reverses for large NOE magnitude exceeding 10 ppb hour$^{-1}$, reflecting that NOE events may have
larger influences on ozone diurnal cycle in the northern China (Figure 3).

Figure 4 shows the time distribution of NOE events (i.e. the hour when the maximal $\Delta O_3/\Delta t$ occurs) in the five key cities
(Beijing, Shanghai, Guangzhou, Chengdu, and Urumqi) representative for different regions in China (Figure 2a), and compares

the time series of ozone, NO$_2$, and CO in the NOE events versus the NNOE events. Each city contains a number of monitoring
sites, and we average the data across the sites within the city to represent ozone at the city level. We identify 211-419 NOE
events for the five cities in the warm season during the 2014-2019 period, corresponding to the frequency of 19%-38%, and
66-314 NNOE events corresponding to the frequency of 6%-29%. The estimated NOE frequency at city level based on the
site-average ozone values is lower than that at site level as the occurrence time of NOE events may vary at different sites.

We find that the timing of the NOE event occurrence shows different patterns at these five cities, however, all cities consistently
show a small fractions of NOE event occurrence at early night (20:00-21:00 LT). This is reasonable because the decreasing
rate of ozone is fast at early night, due to the rapid chemical loss through NO$_x$ titration (NO+O$_3$→NO$_2$+O$_2$) when emissions
are still active and NO levels are high, as evident by the sharp increase in NO$_2$ with ozone decreases in early night. The ozone

decreasing rate (and also NO$_2$ increasing rate) then slows down with reduced NO levels. In Beijing, the timing of NOE events
is evenly distributed across 0:00-6:00 LT, resulting in a flat ozone change when averaging the ozone time series in all NOE
events (Figure 4a). In Shanghai, Guangzhou, Chengdu, and Urumqi, a slightly larger fraction of NOE events is occurring at
0:00-3:00 LT, shaping the peak ozone values at around 3:00 LT on average, but in general the timings of NOE events are
widely distributed over night (Figure 4b,-4e). The wide distribution in the timing of NOE events indicates that NOE events are

more likely driven by episodic synoptic process, rather than anthropogenic-driven factors which otherwise should show a
clearer temporal pattern.

Figure 4 illustrates that the occurrences of NOE events change the ozone diurnal cycle and lead to a secondary ozone peak at
night. It also reveals the distinguished differences in the temporal evolution of NO$_2$ and CO between the NOE and NNOE

events. NO$_2$ and CO typically shows increases in the early night because of the shrinking planetary boundary layer, continuous
nighttime emissions, and titration (for NO$_2$ only), and peak at most cities at around 22:00 LT. The concentration of both species
decreases afterwards, due to the combined effect of chemical destruction (NO$_2$+O$_3$→NO$_3$+O$_2$, NO$_3$+NO$_2$+M→N$_2$O$_5$+M),
mixing, and deposition, though in Shanghai and Guangzhou a midnight peak of CO at around 2:00 LT is observed that may
be linked to regional traffic of heavy vehicles. In particular, we find that NO$_2$ and CO concentrations decrease at a much faster

rate from 22:00-5:00 LT during NOE events, compared to those during NNOE events. This feature has important implications
for understanding the potential driver of NOE events as will be discussed in Section 3.2.



## 3.2 Frequent NOE events associated with high ozone in nighttime residual layer and enhanced vertical mixing

Section 3.1 overviews the general characteristics of NOE events over China, including the spatial and seasonal distributions of NOE event frequencies, peak ozone values, and magnitudes, on the basis of six-year nationwide observations. We now examine what factors determine the overall pattern of NOE event frequencies as displayed in Figure 1a, and trigger the occurrence of NOE events to explain the evolution of ozone, $NO_2$, and CO as shown in Figure 4. Section 3.3 will present typical episodic NOE events as case studies.

The nighttime ozone decreases are due to the combined effect of net chemical loss, as a result of null photochemical ozone production in the absence of sunlight and the ozone depletion by $NO_x$ titration effect, and the continued ozone dry deposition to surface. Reduced ozone loss by weakening the titration effect and deposition can slow down the nighttime ozone decreasing rate but would not increase ozone. Enhancement of nighttime ozone at a given location thus requires external ozone source, either through horizontal transport from adjacent regions with high ozone levels, or through vertical transport/mixing of ozone-rich air to surface. The sharper decreases of $NO_2$ and CO concentrations during NOE events than the NNOE events, as shown in Figure 4, suggest that vertical mixing of air above the surface layer with typically low $NO_2$ and CO levels is a prioritized candidate. This motivates us to examine:

(1) whether the regions with high frequency of NOE events, as shown in Figure 1a, are associated with high concentration of nighttime ozone at the residual layer (the region remaining between the stable nocturnal boundary layer at surface and the free atmosphere), and

(2) whether atmospheric mixing is more active in NOE events that allows the mixing of high-concentration ozone in the residual layer to the surface, compared to NNOE events.

However, examining the first factor is particularly challenged by the extreme sparsity of direct ozone measurements of nighttime residual layer.

Here we propose to use surface ozone averaged over the afternoon (14:00-17:00 LT) as an indicator of ozone level in the nighttime residual layer at a given site. This is because under typical fair-weather conditions, surface ozone is expected to be mixed into the well-developed boundary mixing layer by active atmospheric turbulence in the afternoon. With the sunset, the turbulent mixing in the boundary layer weakens in the absence of solar heating, and the consequent radiative cooling of the ground surface throughout the night forms a stable nocturnal boundary layer with a typical depth of 100-500m above the surface. The erstwhile mixing layer from the daytime, overlying the nighttime boundary layer, is known as the residual layer. It serves as a reservoir for ozone where ozone is not effectively titrated and deposited, so that ozone before sunset is largely remained in the nighttime residual layer (Caputi et al., 2019). Using the afternoon ozone to indicate ozone level in the nighttime residual layer neglects the impact of external transport on ozone to the residual layer, so we suggest that this method makes more sense when the analyses are conducted based on the ensemble of six-year observations, and is achievable for assessing





ozone in nighttime residual layer at thousands of sites as direct measurement of ozone vertical profile is rather sparse. Similarly we may use the surface nighttime $O_x$ ($O_3$+$NO_2$) concentration as an indicator of ozone in the nighttime residual layer (Kleinman et al., 2002; Wang et al., 2018; Tan et al., 2021), assuming that nighttime NO emissions are low and ozone changes due to nighttime titration is much stronger than the effect of deposition and regional transport.

Figure 1b presents the mean afternoon ozone concentrations over China, Europe, and the US. Comparison of Figure 1a and 1b reveals that the pattern of NOE event frequencies and afternoon ozone levels (indicative of ozone in the nighttime residual level) are closely related over China, Europe, and the US, respectively. In particular, sites over China, Mediterranean, and mountainous western US with frequent NOE event recorded are consistently observing high afternoon ozone levels. We also find similar strong correlations between the pattern of nighttime mean $O_x$ level and NOE event frequencies (Figure S4).

High frequency of NOE events (>50%) and nighttime ozone level in the residual layer as indicated by afternoon ozone levels (>50 ppbv) are both found in the industrialized regions over China, including the NCP and YRD city clusters. At night, surface ozone is effectively removed by the $NO_x$ titration and dry deposition, and nighttime ozone levels are on averaged 24 ppbv lower than that in the afternoon (Figure 1c). The afternoon vs nighttime ozone difference is even larger (~30 ppbv) over the

NCP, YRD, and PRD regions. Large ozone contrast between the nighttime residual layer and the boundary layer is expected as the afternoon ozone can be largely remained in the nighttime residual layer where ozone is not effectively destructed and deposited. This is evident by a number of measurements of nightime vertical ozone profiles indicating large ozone gradient between the surface and 100-300 meters aloft at different regions in China (Chi et al., 2018; Wang et al. 2018; He et al., 2021). The ozone-rich air in the residual layer can be effectively mixed to the ground once atmospheric mixing is triggered or

enhanced by synoptic processes. Regions with higher nighttime ozone in the residual layer are more likely to experience surface ozone increase exceeding the threshold of NOE event definition of $\Delta O_3/\Delta t > 5\ ppb\ hour^{-1}$, compared to regions with lower ozone at similar frequency of enhanced nighttime mixing. This explains the observed co-location of high afternoon ozone concentrations and NOE event frequencies as shown in Figure 1a and 1b. The strong relationship between the two also explains why the seasonal shifts in frequency and magnitude of NOE event are similar to those in ozone pollution level as

presented Section 3.1.

We also see equivalent high frequency of NOE events in Mediterranean and western US, with the afternoon ozone levels (~40 ppbv) sufficiently high compared to other sites in the Europe and US but lower compared to those over China. The afternoon vs nighttime ozone contrast over the Mediterranean and western US is much smaller compared to those observed over China.

Ozone at the high-elevation sites over the Mediterranean and western US are largely contributed by high background ozone and frequent downward mixing, while local anthropogenic influences are relatively small (Zhang et al., 2014; Dayan et al., 2017; Jaffe et al., 2018). Frequent downward subsidence of ozone-rich air in the free troposphere that may occur at any time of the day can bring external ozone input to the mountainous surface, resulting in high NOE frequencies there. In other regions





over Europe and US, we see much lower NOE event frequencies on average because nighttime ozone concentrations in the
residual layer (as indicated by the afternoon ozone at surface of 30 ppbv or less) are typically low, which cannot serve as an
effective source to enhance ozone at the surface even there is strong mixing or transport.

Our analyses above show that the high ozone levels in the nighttime residual layer that largely remained from daytime ozone
provide a critical source of nighttime ozone enhancement at the surface. The trigger of a specific NOE event requires the
mixing of ozone-rich air from nighttime residual layer to surface, or horizontal transport of ozone-rich air from adjacent regions.
Active nighttime transport and vertical mixing of air is associated with increasing atmospheric instability, which can be
quantitatively indicated by friction velocity (U*), planetary boundary layer height (PBLH), and vertical profiles of temperature
(Su et al., 2018; Arrillaga et al., 2019; Wang et al., 2019; Shao et al., 2020). We now evaluate the behavior of atmospheric
instability and the strength of vertical mixing during in the NOE events versus NNOE events using these physical parameters
obtained from the ERA5 dataset (Section 2.3).

Figure 5a and 5c compares the evolution of U* and PBLH at the five representative cities averaged over the NOE events to
that over the NNOE events. Increasing U* and PBLH levels typically indicate a more unstable boundary layer and thus stronger
atmospheric mixing. We do not find significant differences in the absolute values of U* and PBLH between the NOE and
NNOE events (Figure S5), however, we see distinct differences in their temporal evolution. U* and PBLH typically show a
steady decreasing trend throughout the nighttime during NNOE events, while in the NOE events the U* and PBLH show
increase in at least a certain part of the nighttime period (Figure 5a and 5c).

We then calculate the frequency of U* and PBLH increase within two hours before the occurrence time of an NOE event to
see whether the increases in FV and PBLH are consistently found in most of the NOE events across all sites. Using two hours
instead of one is because the response of ozone can be lagged behind the enhanced atmospheric mixing. As shown in Figure
5b and Figure 5d, 70% (65%) of the NOE events are associated with an increase in U* (PBLH) averaged at all 814 sites in
2014-2019, supporting that increasing atmospheric instability is critical for the trigger of NOE events. The enhanced
atmospheric instability, as indicated by the rise in U* and PBLH values, not only enhance the activity of turbulence and vertical
mixing that promotes the downward mixing of ozone-rich air mass from the residual layer to surface, but also dilutes the
concentration of $NO_x$ and weaken the titration effect. The combine effects of mixing-induced ozone enhancement and
weakened titration-induced ozone loss increase the probability of NOE events. This also explains the sharper decrease in $NO_2$
and CO during NOE events than the NNOE events as shown in Figure 4, as $NO_2$ and CO concentrations are much lower in the
nighttime residual layer than at the surface.

The enhanced atmospheric instability and vertical mixing in the NOE events can be further supported by the differences in the
evolution of vertical temperature profiles during the NOE events versus NNOE events, as shown in Figure 6. During the NOE



events, at all the five representative cities, the decreases in temperature with time driven by long-wave radiation cooling extend from the ground to 700 hPa or higher, reflecting effective heat exchange between the land and near-surface atmosphere and

295 thus more active mixing. In comparison, the temperature decreases are largely limited in the lowest 100 hPa during the NNOE events, indicating stable and shallow nighttime boundary layer which is not favorable for mixing of air pollutants.

Previous studies also showed that horizontal transport of ozone-rich air from polluted region can lead to episodic NOE events (Sousa et al., 2011; Ghosh et al., 2013; He et al., 2021). We expect that horizontal transport contributes to ozone enhancement

in a specific area if there is higher ozone level upwind. The ozone transport efficiency and magnitude are dependent on the ozone gradient between the source and receptor, and the transport pathway and distance that is closely relevant to the rapid-shift weather conditions. In addition, efficient horizontal transport is typically driven by synoptic processes that may also involve enhanced vertical mixing. These characteristics make it hard to classify the contribution of horizontal transport to NOE events at the national scale, but we will explore their role in the case studies below.

**3.3 Case studies of NOE events**

The above analyses have illustrated that the high afternoon ozone produced from intensive anthropogenic emissions provide rich ozone in the nighttime residual layer, leading to the overall higher frequency of NOE events in China especially in the industrialized regions compared to other regions, and the enhanced mixing of ozone-rich air in the nighttime residual layer to the surface is a critical mechanism to trigger NOE events. We now present three cases of NOE events at Beijing (representative

of the NCP region) and Guangzhou (representative of the PRD region) to zoom in what specific synoptic processes may contribute to the increase in vertical mixing and trigger NOE events.

Figure 7 shows an NOE event observed on July 29, 2015 at multiple sites in Beijing triggered by a convective system. Ozone levels in Beijing exceeded 120 ppbv in the afternoon, decreased rapidly with the sunset, and started increase at around 23:00

LT. The magnitude of this NOE event was up to 36 ppbv with the nighttime peak value exceeding 70 ppbv.

We find that at all sites the nocturnal ozone increases were accompanied with sharp decrease in $NO_2$ and CO concentrations and an increase in PBLH, a typical signal of enhanced vertical mixing from the ozone-rich air in the residual layer. The large hourly precipitation of 8.4 mm at 23:00 LT indicates that there was a convective storm with the occurrence of NOE event, and

320 we also find a follow-up decrease in the equivalent potential temperature ($\theta_{se}$, derived from temperature and relative humidity), a typical feature of the downdrafts in convective storms (Figure 7d) (Jia et al., 2015; Zhu et al., 2020). Examination of the large-scale synoptic pattern suggests that this convective storm was triggered by a cold vortex. Beijing was located in the periphery of subtropical high at 500 hPa, and was strongly influenced by southwest warm-wet flow which transported abundant water vapor to Beijing (Figure S6). Analyses of vertical velocity in Figure 8 show that there was strong upward movement

before 23:00 LT, lifting warm and moist air and facilitating the formation of convective storms. With the start of rainfall, the





precipitation particles fell into the dry air below, and downdrafts were strengthened via evaporative cooling, motivated the transport of ozone from upper layers to surface. The case thus illustrates that strengthened downdrafts and vertical mixing triggered by convective storms is a driver of NOE events. Jia et al. (2015) and Zhu et al. (2020) also presented cases of convective storm-induced NOE events with similar evolution of $NO_2$, CO, and meteorological parameters in summer in the NCP region.

Figure 9 shows a case of NOE event under the influence of boundary-layer low-level jets in Beijing on August 26, 2017. Nocturnal ozone concentration started increase at 21:00 LT, flatted at 23:00 LT, and increased again at 02:00 LT. The magnitude of the NOE event was 22 ppb hour$^{-1}$. In this NOE case the nighttime peak ozone concentrations (about 50 ppbv) were even higher than the afternoon ozone concentrations (30 ppbv), suggesting that there was external ozone source besides the daytime ozone remaining in the nighttime residual layer. We find a subsidence branch of air above the Beijing city in this event which may bring ozone-rich air from the middle troposphere to lower troposphere (Figure 10).

Both periods with ozone increases in this event were associated with enhanced PBLH and U* values and declined $NO_2$ and CO levels, again indicating enhanced vertical mixing from air in the upper layer to the surface. Analyses of wind profile show that the wind speed continued increasing and exceeded 10 m s$^{-1}$ in the lower troposphere, shaping a strong vertical wind shear, a typical feature of low-level jets (Figure 9e and Figure S7). The low-level jets and associated wind shear have been shown to produce turbulent kinetic energy, weaken the decoupling of the residual layer and the stable nighttime boundary layer, and aggravate vertical mixing (Banta et al., 2003; Tsiringakis et al., 2022) . We find that the first period of ozone increase in 21:00-23:00 LT was associated with the strengthening of the low-level jets as indicated by the rapid increase in the wind speed from 10 to 13 m s$^{-1}$. There were little increases in wind speed from 23:00-01:00, and correspondently no significant ozone increases in this period. The second period of ozone increases was again with the strengthening of low-level jets with wind speed exceeding 20 m s$^{-1}$ from 800 hPa to 975 hPa (Figure 9e). This case thus presents how the development of boundary layer low-level jets triggers NOE events. The low-level jets have also been identified as a key factor causing multiple NOE events in different regions in the NCP region (Zhu et al., 2020), US (Kuang et al., 2011; Hu et al., 2013; Sullivan et al., 2017; Caputi et al., 2019), and Europe (Kulkarni et al., 2016; Klein et al., 2019).

Figure 11 also illustrates a low-level jets induced NOE event in Guangzhou in the southern China on September 21th, 2014. The afternoon ozone concentrations in Guangzhou reached about 60 ppbv. The nocturnal ozone exhibited continuously increases from 21:00 LT to 04:00 LT with increases in PBLH and decrease in $NO_2$ concentration, lifting surface ozone level from 9 ppbv to 34 ppbv. We find clear development of the low-level jets and wind shear from the wind profiles that can enhance atmospheric mixing as discussed above (Figure 11b).





We further notice that the nocturnal ozone enhancement in this case was not only limited in Guangzhou city, but was instead
spread at multiple sites in the PRD region at a background of strong northerly winds produced by the typhoon "Fung-wong"
(Figure 11c). "Fung-wong" is located in the east of the PRD region, and the PRD region is exposed to the northerly wind of
its circulation (Figure S8). The nocturnal ozone started increase at sites north to Guangzhou city from 21:00 LT, and gradually
propagated to Guangzhou and the southern sites afterward. Figure 1c indicates that the mean nighttime ozone concentrations
are typically below 25 ppbv in the PRD and its northern surrounding regions. However, we find that in this specific case ozone
concentrations at sites north to the PRD region reach 40 ppbv, much higher than those over the PRD region (Figure 11d). The
northerly winds in the background of typhoon circulation are expected to transport the ozone-rich air in the north to the PRD
region, contributing to nocturnal ozone enhancement there. This is also supported by the backward trajectory analyses using
the Hybrid Single Particle Lagrangian Integrated Trajectory (HYSPLIT) model (Figure 11d). The analysis above thus
illustrates that horizontal transport may also contribute to in the NOE events, quantifying the relative contribution from vertical
mixing versus horizontal transport to ozone enhancement would require modelling studies..

### 3.4 Implications for ozone evolution in the next day

We also examine whether NOE events would have clear instruction to predict daytime ozone of the next day in China. Figure
4 shows that in the five representative cities, the occurrence of NOE events tends to induce higher ozone in the early morning
in the next day, compared to the NNOE events. However, it does not necessarily result in higher daytime ozone compared to
the precedent day. We find that the frequency of daytime ozone increase or decrease in the day following an NOE event is
roughly equivalent (46% vs 54%) for all sites in China. Previous studies have shown that ozone in the nighttime residual layer
influenced surface ozone levels in the following day by fumigation of ozone in the residual layer into the developing daytime
boundary layer. While the enhanced nocturnal mixing between the residual layer and nighttime boundary layer contribute to
nocturnal ozone enhancement at the surface, the enhanced ozone is also subject to more efficient chemical destruction and dry
deposition, resulting in lower ozone peak values on the next day (Hu et al., 2013; Caputi et al., 2019). Analyzing the implication
of NOE events in the next day's ozone level will need to further separate the daily variation of ozone from shifts in
meteorological conditions and the resulting impacts on ozone chemistry and transport. This would be an important topic of
ozone air quality research over China in the future.

### 4 Conclusion

In this study, we report the previously unrecognized high frequency of the NOE events in China, present their statistical
characteristics, and explore the possible mechanisms based on six-year (2014-2019) observations from the Chinese national
monitoring network. We find that the annual mean frequency of NOE events is 41±10% averaged over all 814 Chinese sites
in 2014-2019, which is 46% larger than those over Europe and US. Sites with high frequency (>50%) of NOE events are
concentrated in industrialized city clusters (NCP, YRD, and PRD) and in the western China. The NOE frequency is higher in



the warm (46%) than the cold season (36%), consistent with the seasonal evolution of ozone levels. The mean ozone peak during NOE events in the warm season is 37±6 ppbv (31±6 ppbv in the cold season), significantly higher than those in the NNOE events of 17±11 ppbv (10± 9 ppbv), and is sufficiently high to pose negative impacts on human health and vegetation activity. In about 85% of the NOE events the maximum ozone enhancement during NOE events is within 5-15 ppbv hour$^{-1}$, but in 10% of the cases the ozone increase can exceed 20 ppbv hour$^{-1}$. We also find much faster decreasing rate of nighttime $NO_2$ and CO concentrations and increases in U* and PBLH that are indicative of enhanced atmospheric vertical mixing during NOE events compared to NNOE events.

We propose that the high afternoon ozone provides rich ozone source in the nighttime residual layer, determining the overall higher frequency of NOE events at regions with severe ozone pollution than cleaner regions, and then the enhanced atmospheric mixing trigger NOE events by allowing the ozone-rich air in the residual layer to mix into the nighttime boundary layer. Figure 12 illustrates the conceptual model of the formation of the high NOE frequency over China. High ozone is generated by active photochemistry in the daytime, and has relatively small vertical gradient in the daytime boundary layer with effective vertical mixing and turbulence. At night, the surface ozone decreases significantly due to titration in high $NO_x$ environment and dry deposition with the shallowing of nighttime boundary layer, while relatively higher ozone remains in the residual layer. Synoptic processes such as convective storms and low-level jets produce turbulent kinetic energy, weaken the decoupling of the residual layer and the nighttime boundary layer, and aggravate vertical mixing of ozone-rich air in the residual layer into the nighttime boundary layer and cause the NOE event. This is supported by our analyses that more than 70% (65%) of the NOE events are associated with increases in U* and PBLH values, and also supported by the observed sharper decreases in $NO_2$ and CO concentrations in NOE events compared to the NNOE events. In addition, horizontal transport of ozone-rich plumes may also be a supplementary driver of NOE event in areas such as the PRD regions.

Our study thus provides a first overview of the NOE events over China from characteristics to mechanisms. Nevertheless, as we focus more on the general behaviors in NOE events based on six-year observations at hundreds of sites than episodic cases, the proposed conceptual diagram of the NOE event mechanism may not cover all the NOE events. We call for more direct observations of vertical structure of ozone and its evolution from daytime to nighttime (Kuang et al., 2011; Jia et al., 2015; Caputi et al., 2019; He et al., 2021), and more 3-D chemical modelling studies (Hu et al., 2013; Klein et al., 2014) to better explore the contribution of mixing and regional transport to NOE events, and to further analyze the impacts of NOE events on atmospheric chemistry, human health, and vegetation productivity.

**Data availability.** Hourly observational air pollutants are available at http://106.37.208.233:20035/ for China, https://aqs.epa.gov/aqsweb/airdata/download files.html for the United States, and https://discomap.eea.europa.eu/Index/ for Europe. The ERA5 reanalysis data is accessed via https://cds.climate.copernicus.eu/#!/home.





**Author contributions.** XL and SJF designed the study. CH conducted the data collection and analyses with contributions
from HLW, HCW, YL, GWH, YPH, YRW, YLZ, YML, and QF. All authors provided practical comments. CH, XL, and SJF
wrote the paper with input from all authors.

**Competing interests.** The authors declare that they have no conflict of interest.

**Acknowledgments.** The authors thank all the data contributors and the NOAA ARL for the provision of the HYSPLIT
transport and dispersion model utilized in this study.

**Financial support.** This research has been supported by the Key-Area Research and Development Program of Guangdong
Province (grant no. 2020B1111360003), Guangdong Major Project of Basic and Applied Basic Research (grant no.
2020B0301030004), the Guangdong science and technology plan project (grant no. 2019B121201002), the National Natural
Science Foundation of China (NSFC, grant no. 41030164), and the Guangdong Basic and Applied Basic Research Foundation
(grant no. 2022A1515011554) .

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





**Figure 1.** Comparison of annual frequency of nocturnal ozone enhancement (NOE) event, mean afternoon (14:00-17:00 LT) ozone, and mean nighttime (20:00-06:00 LT) ozone between China (CH), Europe (EU), and the United States (US), averaged over 2014-2019. Only sites with continuous measurements are included. The regional mean and standard deviations among the N sites are shown inset.






**Figure 2.** Comparison of the NOE event frequency and nocturnal peak ozone concentrations between the warm season (April-September, panels a and c) and cold season (October-March, panels b and d) in China. The regional mean and standard deviations among the N sites are shown inset.



Frequency of NOE event magnitude (maximum ΔO₃/Δt)



**Figure 3.** Frequency of different magnitudes of the NOE events (the maximum of $\Delta O_3/\Delta t$ in an NOE event) in the warm season in China. The regional mean and standard deviations among the N sites are shown inset.





**Figure 4.** Comparison of the temporal variation of nocturnal air pollutants (ozone, NO₂, and CO) between the NOE and the NNOE events in the warm season in five typical cities (Beijing, Shanghai, Guangzhou, Chengdu, and Urumqi). Each city contains a number of monitoring sites, and we average the data across the sites within the city to represent ozone at the city level. The number in parentheses represents the frequency of NOE and NNOE event in the city. The estimated NOE frequency at city level based on the site-average ozone values is lower than that at site level as the occurrence time of NOE events may vary at different sites. The pie chart shows the frequency of the occurrence time of the NOE events, defined as the time with maximum ($\Delta O_3 / \Delta t$) in each NOE event.

**Figure 5.** Increases in friction velocity (U*) and planetary boundary layer height (PBLH) in the NOE event. Panels (a) and (c) shows the nighttime evolution of the U* and PBLH, scaled by the values at 20:00 LT, averaged over all NOE and NNOE events in the five representative cities. The absolute values of U* and PBLH are shown in Figure S5. Panels (b) and (d) show the frequency of U* and PBLH increase within two hours before the occurrence time of an NOE event at individual sites. The regional mean and standard deviations among the N sites are shown inset.



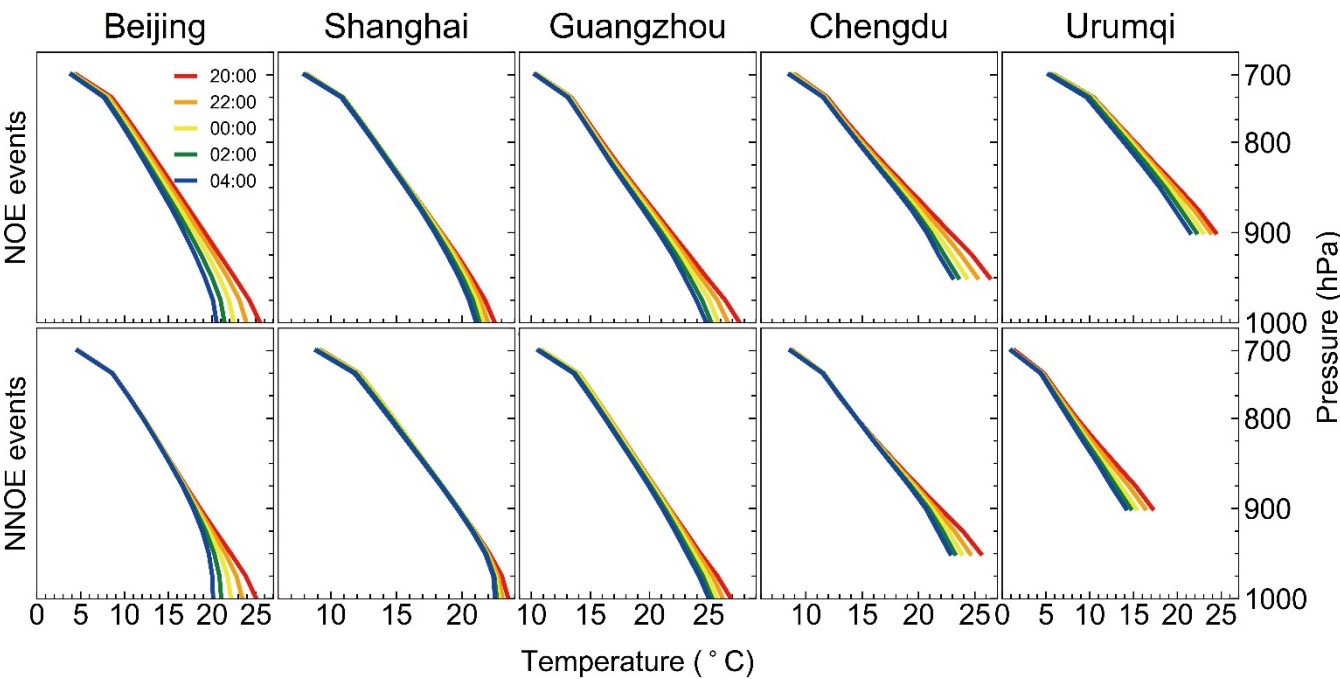

**Figure 6.** Comparison of vertical profiles of temperature at five typical cities between the NOE (the top panels) and the NNOE (the bottom panels) events. The colored lines represent temperature profiles at different time of the night.

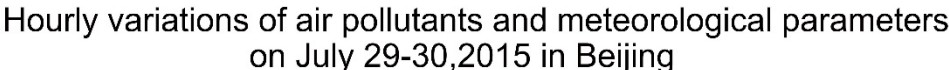

**Figure 7.** Hourly variations of air pollutants and meteorological parameters at four sites in a Beijing NOE event on July 29-30, 2015, induced by a convective storm. Panels (a), (b), and (c) show the evolution of ozone, NO₂, and CO, respectively. Panel (d) shows the evolution of PBLH, equivalent potential temperature (θse), and precipitation from the ERA5 dataset. The vertical dashed line marks the start of nighttime.


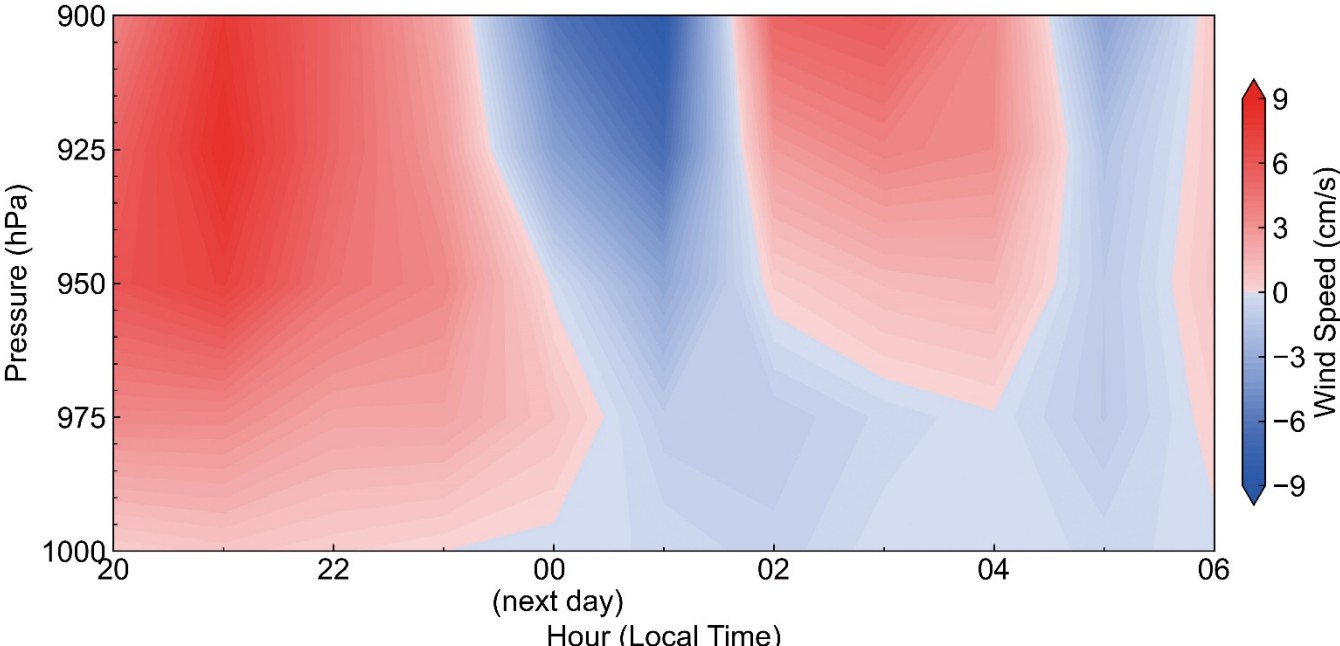

**Figure 8.** Evolution of vertical velocity over Beijing (116.5ºE, 40º N) on July 29-30, 2015 in an NOE event.



**Figure 9.** Hourly variations of air pollutants and meteorological parameters at four sites in a Beijing NOE event on August 26-27, 2017, induced by low-level jets. Panels (a), (b), and (c) show the evolution of ozone, $NO_2$, and CO, respectively. Panel (d) shows the evolution of PBLH and frequent velocity ($U^*$). Panel (e) shows the temporal variation of the horizontal wind profiles, with the color showing the wind speed. The vertical dashed line marks the start of nighttime.





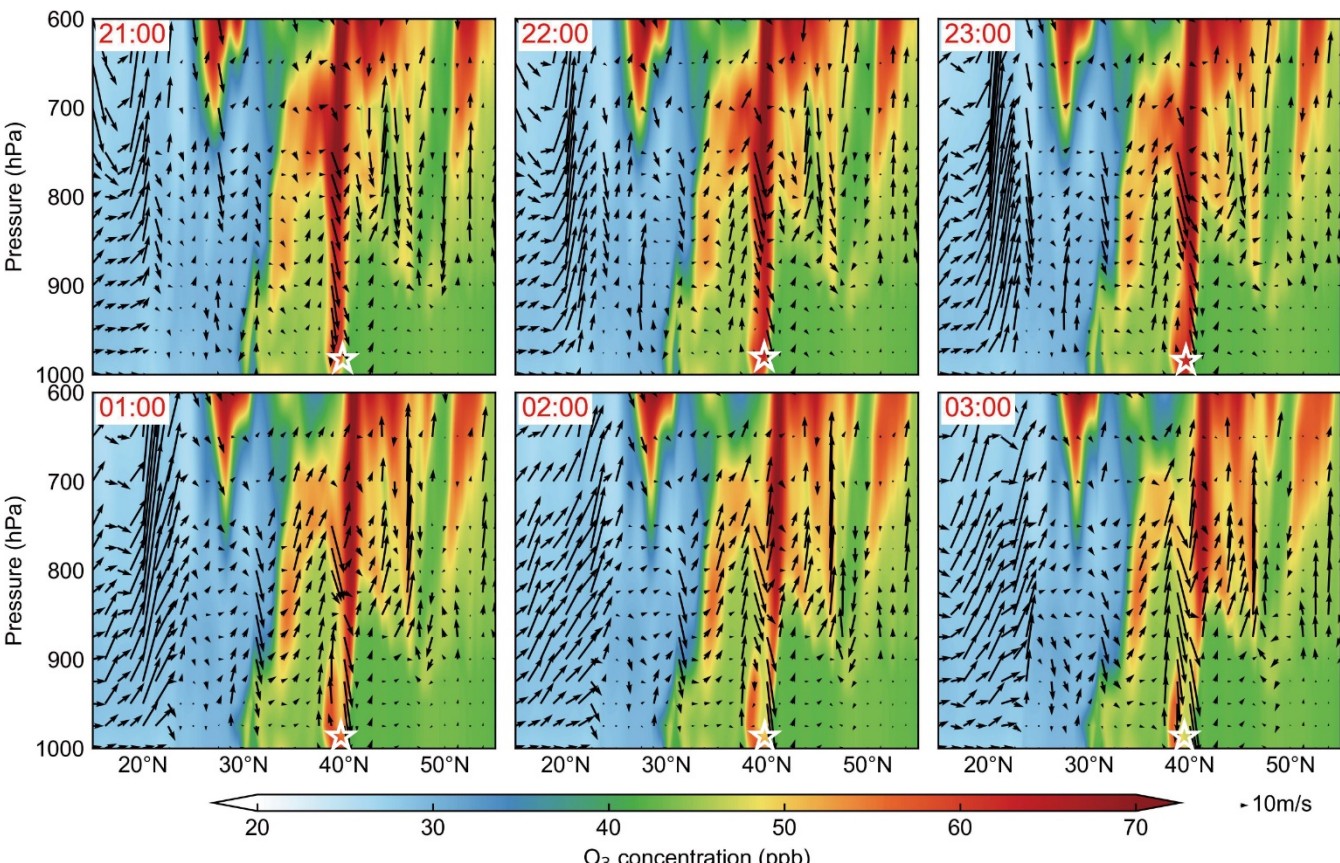

**Figure 10.** Latitude-altitude cross-section of ozone (shaded) and wind fields of meridional (south-north) and vertical wind (scale by $10^3$) (arrows) at the longitude of 116.5°E in a Beijing NOE event on August 26-27, 2017. The star marks the location of Beijing.



**Figure 11.** An NOE event in Guangzhou on September 21-22, 2014. Panel (a) shows the hourly variations of nocturnal ozone, NO₂, and PBLH. Panel (b) shows the evolution of vertical profiles of wind speed. Panel (c) shows the 10 m wind fields and changes in ozone relative to the previous hour. Panel (d) shows the 10 m wind fields and hourly ozone concentration. The purple line is the 12-h backward trajectories of air mass calculated using the HYSPLIT model.





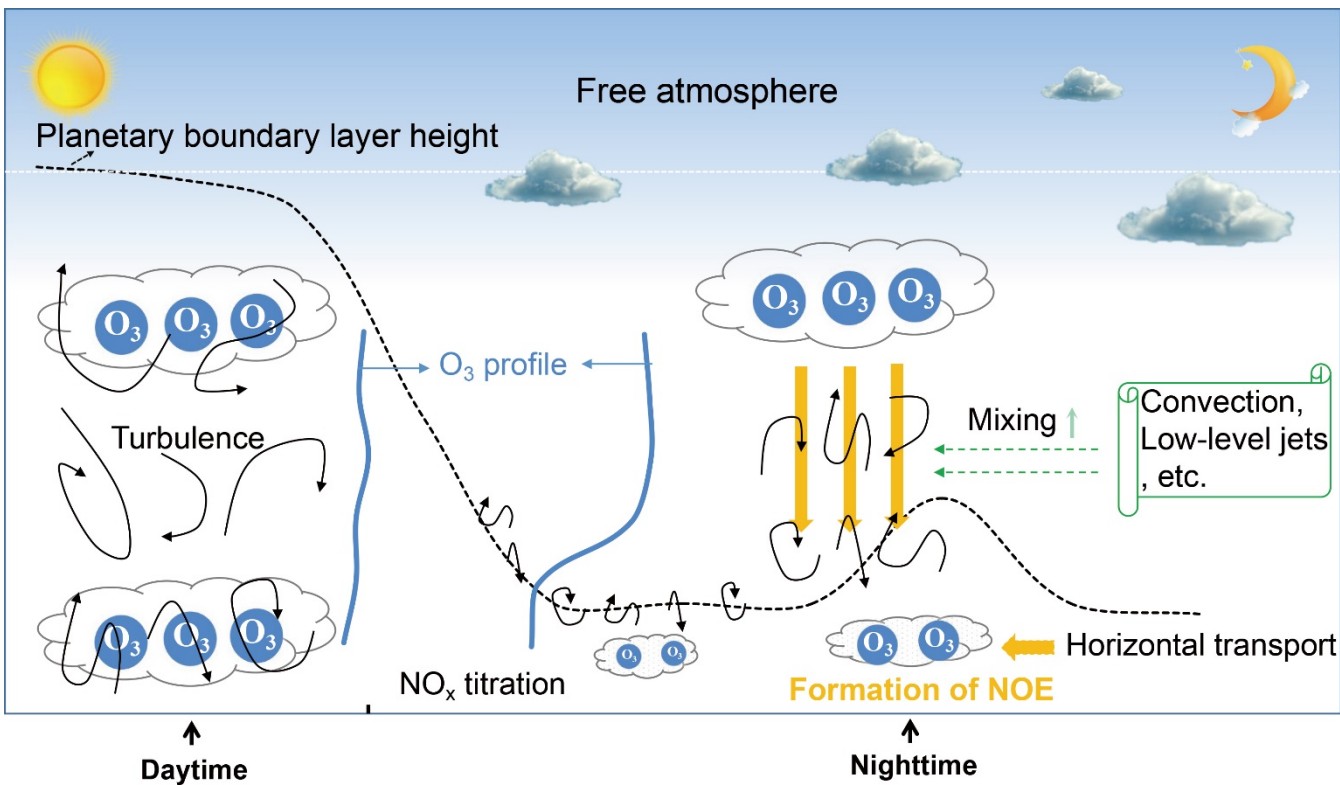

**Figure 12.** Conceptual model of high frequency of NOE events over China.