# Peer review of "Unexpected high frequency of nocturnal surface ozone enhancement events over China: Characteristics and mechanisms"

_Atmospheric Chemistry and Physics, 2022_

## Author Comment (AC1)

**Reviewer #1**

**Comment [1-1]:** GENERAL: This paper focuses on the characteristics and mechanisms of nocturnal ozone enhancement (NOE) events. Cases with surface ozone enhancement of 5 ppbv/hour or greater in one of any two adjacent hours in 20:00-06:00 LT are defined as NOE events. Frequencies of NOE events are calculated for 814 sites in China, 762 sites in the US and 1880 sites in EU countries in 2014-2019. The annual frequency of NOE events over China is found unexpectedly high (41%+/-10%) and much higher than those over the US and EU. Higher afternoon ozone levels (as proxies of nocturnal ozone levels in the residual layer) are believed to be the precondition of NOE events. It is confirmed by cases studies that the NOE events in the surface layer is triggered by enhanced atmospheric mixing during processes like convective storms and low-level jets. More NOE events are found in warm season than in cold season. Distributions of NOE events of different magnitudes over China are presented and discussed as well as the timing of NOE events and nighttime variations of ozone, $NO_2$, CO, friction velocity and PBLH in NOE and non-NOE events in five Chinese cities.

NOE events have been found at some sites in different parts of the world and reported in the literature. Previous studies have already shown that the NOE events are caused mainly by convective storms, low-level jets, horizontal transport, etc. To the best of my knowledge, however, there has been no previous publication presenting nationwide statistics of NOE events over China or the comparison of NOE events over China with those over EU and the US. In this sense, this paper is original and within the scope of ACP. The methods applied in this paper are mostly valid. The results presented are interesting and generally sound. The paper is well structured and written. It can be improved by appropriately addressing the following issues. I recommend publication of this paper in ACP after revisions.

**Response [1-1]: We thank the reviewer for the positive and valuable comments. All of them have been implemented in the revised manuscript. Please see our itemized responses below.**

**Comment [1-2]:** MAJOR COMMENTS: In this study, a NOE event is defined as ozone increase by at least 5 ppbv/hour in one of any two adjacent hours in 20:00-06:00 LT. The selection of the threshold (5 ppbv/hour) for NOE seems to be arbitrary. As the threshold value substantially impacts not only the statistics of NOE events but also the results like the contrasts between regions and between warm and cold seasons, it should be determined based on scientific analysis and consideration. The observations of ozone and also other species are always fluctuating in a certain degree due to factors like turbulences, source/sink disturbances, transport, etc. The intensities of fluctuations related to different factors should vary in a large range and may be dependent of season and location. I think you may obtain a kind of fluctuation intensity spectrum for each site by plotting the frequencies against the delta[$O_3$]/delta(t) values. I do not know how the spectrum may look like but guess it

might not be monotonic. If the spectrum is really not a monotonic curve, you may relatively easily determine your threshold based on your scientific considerations. Otherwise it might be difficult for you to determine the threshold and convince the readers of your threshold. I think the focus of this paper is the NOE event that is really caused by any particular atmospheric condition or process. The nocturnal ozone fluctuations occur daily under normal atmospheric conditions should not be included in the NOE statistics. In particular, when you are using "unexpected high frequency of" NOE in your title, the threshold definition must be supported by scientific analysis.

**Response [1-2]: Thank you for pointing it out. We were hoping to be consistent with previous studies (Eliasson et al., 2003; Zhu et al., 2020) by defining the NOE event as ozone increase by >5 ppb hour$^{-1}$ in one of any two adjacent hours in 20:00-06:00 local time. We agree that this threshold should be carefully examined.**

**We have followed your suggestion to derive the fluctuation intensity spectrum for each site by plotting the frequencies of the $\Delta O_3/\Delta t$ values. Results are shown in Figure R1. We find that the nocturnal ozone $\Delta O_3/\Delta t$ values generally follow the Gaussian distribution, based on observations at all Chinese sites in 2014-2019. About 70% of the $\Delta O_3/\Delta t$ values are negative, reflecting the expected dominant ozone decrease at nighttime. We find a possibility with $\Delta O_3/\Delta t$ >5 ppb hour$^{-1}$ of 7.7%, which is mostly outside the two-sigma standard deviation at Gaussian distribution, suggesting that $\Delta O_3/\Delta t$ >5 ppb hour$^{-1}$ cases are not likely "normal fluctuation" under normal atmospheric conditions. Using the threshold of $\Delta O_3/\Delta t$ greater than 4 or 6 ppb hour$^{-1}$ does not change the spatial pattern of NOE frequency. We have also tried to define a NOE by considering the relative fluctuation of nocturnal ozone (i.e., nighttime ozone enhancement normalized by the corresponding afternoon ozone level), but we find that it much complicates the analyses by introducing low $\Delta O_3/\Delta t$ values when afternoon ozone level is low, which has weak implication. We thus prefer to stick to this threshold so that our results are comparable to existing studies.**

**We have added the following text in Section 2.3 for justification: "Following previous studies of Eliasson et al. (2003) and Zhu et al. (2020), we define a nocturnal ozone enhancement (NOE) event if ozone concentration at a site increases by more than 5 ppbv ($\Delta O_3/\Delta t > 5$ ppb hour$^{-1}$) in one of any two adjacent hours in the nighttime period. We find that nocturnal $\Delta O_3/\Delta t$ values at Chinese sites generally follow the Gaussian distribution, and $\Delta O_3/\Delta t > 5$ ppb hour$^{-1}$ cases only account for 7.7% of the $\Delta O_3/\Delta t$ dataset, indicating that this definition should have effectively ruled out nocturnal ozone fluctuations occur under normal atmospheric conditions (Figure S2). We only define one NOE event if there are more than one hour with $\Delta O_3/\Delta t > 5\, ppb\, hour^{-1}$ at a specific night, and observations with maximum $\Delta O_3/\Delta t$ are used for statistical analyses."**

[Figure]

**Figure R1 Probability density distribution of hourly nocturnal ozone fluctuation ($\Delta O_3/\Delta t$) at all Chinese sites (represented by each curve) in 2014-2019.**

**Reference:**

Eliasson, I., Thorsson, S., and Andersson-Sköld, Y.: Summer nocturnal ozone maxima in Göteborg, Sweden, Atmospheric Environment, 37, 2615-2627, https://doi.org/10.1016/S1352-2310(03)00205-X, 2003.

Zhu, X., Ma, Z., Li, Z., Wu, J., Guo, H., Yin, X., Ma, X., and Qiao, L.: Impacts of meteorological conditions on nocturnal surface ozone enhancement during the summertime in Beijing, Atmospheric Environment, 225, 117368, https://doi.org/10.1016/j.atmosenv.2020.117368, 2020.

**Comment [1-3]:** The regional and seasonal differences in the NOE frequencies are all impressive. Data show that regions with higher frequencies of NOE events are associated with higher levels of afternoon ozone. However, the real cause of the regional and seasonal differences in the NOE frequencies is not clear. Are the NOE differences caused by the differences in atmospheric processes (convective storm, low-level-jet, etc.) or purely the ozone level differences or both? To answer this question, it is suggested to consider the relative fluctuation of nocturnal ozone (i.e., nighttime ozone enhancement normalized by the corresponding afternoon ozone level) as the metric of a NOE event (again, the threshold should be carefully determined).

**Response [1-3]: Thank you for pointing it out. We have compared in Figure R2 the spatial and seasonal pattern of the NOE frequencies, afternoon ozone level, and the frequency of $\Delta U^*/\Delta t$ ($\Delta PBLH/\Delta t$) enhances by 30% in one of any two**

adjacent hours in the nighttime period, as an indicator of enhanced nighttime atmospheric mixing. We can see that regions with relatively higher frequency of enhanced nighttime U* or PBLH are located in the Sichuan Basin, the Fenwei Plain (Shanxi Province), and Pearl River Delta. Coastal regions typically show lower frequency. This spatial pattern is clearly not consistent with the hotspots of NOE frequency as shown in Figure R2a and 2e, suggesting that the overall regional differences in the NOE frequencies are more likely to be driven by afternoon ozone than enhanced atmospheric mixing. However, for some regions such as the Sichuan Basin, atmospheric processes may be more important as indicated by the high frequency of enhanced $\Delta U^*/\Delta t$ ($\Delta PBLH/\Delta t$). We also find that the frequency of enhanced nighttime atmospheric mixing in warm season is indeed much higher than that in the cold season, indicating that atmospheric processes may also contribute to the seasonal difference of NOE patterns. We now state in the text "We also find higher frequency of nighttime U* and PBLH enhancement in the warm season than that in the cold season, suggesting that seasonal difference in nighttime atmospheric mixing activity also contributes to that in NOE frequency as shown in Figure 2." Here we do not use relative fluctuation of nocturnal ozone as a metric to define NOE. Please kindly find our response in [Response 1-2].

[Figure]

**Figure R2. Comparison of the pattern of the NOE frequencies, afternoon ozone level, and the frequency of $\Delta U^*/\Delta t$ ($\Delta PBLH/\Delta t$) increase by 30% at nighttime period. The values of regional mean and standard deviations among the N sites are shown inset at the top of each figure (mean ± standard deviation).**

**Comment [1-4]:** MINOR CONCERN/EDIT: L110: In the abstract section the NOE event is defined as ozone increase by at least 5 ppbv/hour, meaning equal to or greater than 5 ppbv/hour. This is not consistent with >5 ppbv/hour stated here. In addition, it is not clear which number is counted if two or more cases occur with enhancement

over the threshold during one night. In other word, can the NOE event in a day be more than one?

**Response [1-4]: Sorry for the confusion. We have now clarified in the abstract "(NOE, defined as ozone increase by more than 5 ppbv hour$^{-1}$ in one of any two adjacent hours in 20:00-06:00 local time)" This is now consistent with the statement in Section 2.3. We also state in Section 2.3 "We only define one NOE event if there are more than one hour with $\Delta O_3/\Delta t > 5$ ppb hour$^{-1}$ at a specific night, and the observations with maximum $\Delta O_3/\Delta t$ are used for statistical analyses"**

**Comment [1-5]:** L176: what do you mean by "evenly distributed"? The statistics (Figure 4a) for this time period are 18%, 29%, and 19%.

**Response [1-5]: We have revised the statement in the text to "In Beijing, the timing of NOE events is diversely distributed across 0:00-6:00 LT with a frequency ranging from 18 to 29%, resulting in a flat ozone change when averaging the ozone time series in all NOE events (Figure 4a)"**

**Comment [1-6]:** L274: I think substantial differences in the absolute values of U* and PBLH between the NOE and NNOE events are required if the NOE is really caused by enhanced vertical mixing. The differences may have been masked by averaging effect, average over six years and different sites. Case studies using data from individual sites may make it clear.

**Response [1-6]: We have plotted the absolute values of U* and PBLH between the NOE and NNOE events in Figure S4, and indeed find that the values are significantly higher in NOE than NNOE events for all the selected cities except for Beijing, but the plot is somewhat scattered because the values for each city differ a lot. As we focus more on the relative enhancement of U* and PBLH than mean values, we prefer to present the relative ratio to the 8 p.m. (LT) value in the main text and place the absolute value plot in the supplement. Case studies indeed support the large value and the enhancement of U* and PBLH with NOE event. We have revised the text to "We find that the absolute values of nighttime U* and PBLH are generally larger in NOE than NNOE events (Figure S4). More importantly, we see distinct differences in their temporal evolution. U* and PBLH typically show a steady decreasing trend throughout the nighttime during NNOE events, while the U* and PBLH show increase in at least a certain part of the nighttime period in NOE events, suggesting that atmospheric mixing is becoming more active (Figures 6a and 6c)."**

**Comment [1-7]:** L280: FV or U*? Be consistent.
**Response [1-7]: We have revised FV to U* in the text.**

**Comment [1-8]:** L283: it is worth knowing which process is the most important one that causes the increasing atmospheric instability.
**Response [1-8]: We agree. We attempt to shed some lights on specific processes**

contributing to increasing atmospheric instability in the case studies, but much more work is required to quantify the relative contribution from each process (e.g. LLJs, convections) to the NOE events at different region. We have added the discussion in the Conclusion remark "We call for more direct observations of vertical structure of ozone and its evolution from daytime to nighttime (Kuang et al., 2011; Jia et al., 2015; Caputi et al., 2019; He et al., 2021), and more 3-D chemical modelling studies (Hu et al., 2013; Klein et al., 2014) to quantitatively explore the contribution of mixing and regional transport to NOE events (including the underlying synoptic processes such as low-level jets and convective storms), and to further analyze the impacts of NOE events on atmospheric chemistry, human health, and vegetation productivity. "

**Reference**

Caputi, D. J., Faloona, I., Trousdell, J., Smoot, J., Falk, N., and Conley, S.: Residual layer ozone, mixing, and the nocturnal jet in California's San Joaquin Valley, Atmos. Chem. Phys., 19, 4721-4740, 10.5194/acp-19-4721-2019, 2019.

He, Y., Wang, H., Wang, H., Xu, X., Li, Y., and Fan, S.: Meteorology and topographic influences on nocturnal ozone increase during the summertime over Shaoguan, China, Atmospheric Environment, 256, 118459, https://doi.org/10.1016/j.atmosenv.2021.118459, 2021.

Hu, X.-M., Klein, P. M., Xue, M., Zhang, F., Doughty, D. C., Forkel, R., Joseph, E., and Fuentes, J. D.: Impact of the vertical mixing induced by low-level jets on boundary layer ozone concentration, Atmospheric Environment, 70, 123-130, https://doi.org/10.1016/j.atmosenv.2012.12.046, 2013.

Jia, S., Xu, X., Lin, W., Wang, Y., He, X., and Hualong, Z.: Increased Mixing Ratio of Surface Ozone by Nighttime Convection Process over the North China Plain, J Appl Meteor Sci, - 26, - 280, - 10.11898/1001-7313.20150303, 2015.

Klein, P. M., Hu, X.-M., and Xue, M.: Impacts of Mixing Processes in Nocturnal Atmospheric Boundary Layer on Urban Ozone Concentrations, Boundary-Layer Meteorology, 150, 107-130, 10.1007/s10546-013-9864-4, 2014.

Kuang, S., Newchurch, M. J., Burris, J., Wang, L., Buckley, P. I., Johnson, S., Knupp, K., Huang, G., Phillips, D., and Cantrell, W.: Nocturnal ozone enhancement in the lower troposphere observed by lidar, Atmospheric Environment, 45, 6078-6084, https://doi.org/10.1016/j.atmosenv.2011.07.038, 2011.

**Comment [1-9]:** L359-370: the case with typhoon "Fung-wong" may be more complicated than just transport of ozone-rich air in the north to the PRD region". It is known that typhoon processes may strongly impact the surface ozone level in the periphery of typhoons. Descending air usually play a key role in these processes. Even ozone in the upper troposphere and lower stratosphere can be transported down to the surface (e.g., Jiang et al., Why does surface ozone peak before a typhoon landing in southeast China?, Atmos. Chem. Phys., 15, 13331–13338, https://doi.org/10.5194/acp 15-13331-2015,2015).

**Response [1-9]: Thank you for pointing it out. We have added the following text in Line 394 "Descending air in the periphery of the typhoon can trigger vertical**

**transcript of O₃-rich air from upper troposphere or even lower stratosphere to the surface, contributing to ozone enhancement (Jiang et al., 2015).”**

**Reference**

Jiang, Y. C., Zhao, T. L., Liu, J., Xu, X. D., Tan, C. H., Cheng, X. H., Bi, X. Y., Gan, J. B., You, J. F., and Zhao, S. Z.: Why does surface ozone peak before a typhoon landing in southeast China?, Atmos. Chem. Phys., 15, 13331-13338, 10.5194/acp-15-13331-2015, 2015.

**Comment [1-10]:** L374-375: I think it depends highly on the timing and strength of the NOE event. Of course, it is not so simple considering the variations of ozone precursors and redistribution in the vertical direction.

**Response [1-10]: We agree. We have revised the text: “However, it does not necessarily result in higher daytime ozone compared to the precedent day, which highly depends on the timing and strength of the NOE event and the variations of ozone precursors and redistribution in the vertical direction.”**

**Comment [1-11]:** L390: the seasonal variation of surface ozone in the PRD region is much different from those in other Chinese regions. More of the NOE events in the PRD occur in cold season than in warm season. Perhaps it is better to point out this particularity.

**Response [1-11]: Thank you for pointing it out. We have added the point in text: “The NOE frequency is higher in the warm (46% averaged for all sites) than the cold season (36%) in most regions, except for the PRD region where NOE events occur at higher frequency in the cold season than in the warm season, consistent with the seasonal evolution of ozone levels.”**

**Comment [1-12]:** Figure 6: the temperature differences between NOE and NNOE events are very large. Why?

**Response [1-12]: The differences between NOE and NNOE events in terms of the absolute values are small in all selected cities except for Urumqi. We find that this is because the NOE events in Urumqi are concentrated on May-August when temperature is significantly higher than April and September. In terms of the vertical profiles, the differences between the NOE and NNOE events reflect the more effective heat exchange between the land and near-surface atmosphere with active mixing in the NOE events, as stated in the text. We have now replaced this plot with the potential temperature following the other referee's comment [Response 2-7]. The results are consistent. We have added the following text “The higher potential temperature in Urumqi in the NOE than NNOE events is due to the higher NOE frequency in May-August when temperature is significantly higher than April and September.”**

---

## Author Comment (AC2)

**Reviewer #2**

**Comment [2-1]: Major comments:** This paper presents a large dataset of the dynamics of nocturnal $O_3$ in China with a special emphasis on the frequency with which nighttime concentrations are observed to increase (nocturnal ozone enhancements or NOEs). While I think there are useful results here, the paper would benefit from substantial editing to improve clarity and conciseness. I recommend publication after the following comments have been addressed.

**Response [2-1]: We thank the reviewer for the positive and valuable comments. All of them have been implemented in the revised manuscript. Please see our itemized responses below.**

**Comment [2-2]:** I recommend reframing the motivation behind this study. As written the introduction seems to argue that these NOEs are interesting because of their effects on health (of humans, plants, etc) but, at least in China, the maximum levels observed during NOEs are really very low (17 ppb in winter, 37 ppb in the summer) for health effects, especially as they occur during a time of day when most humans are asleep indoors and plants are more dormant. Do you have references that show those concentrations being associated with negative health outcomes? I don't mean to say at all that we shouldn't try to understand the full daily cycle of $O_3$ and the effects of vertical mixing on nighttime levels, just that it rings hollow to present it as if the NOEs themselves are a major source of concern. I also found it a bit odd that there seems to be more focus given to the frequency of occurrence of increases than to the concentrations themselves.

**Response [2-2]: Thank you for the suggestion. We have reframed the second paragraph of the introduction to highlight the implication of NOE events to atmospheric dynamics and chemistry: "Analyses of the nocturnal ozone enhancement events and associated ozone peaks have important implications for understanding nocturnal atmospheric dynamics and chemistry, as well as ozone exposure to human and vegetation health. Due to the lack of nighttime chemical source, elevated nocturnal ozone levels are thought to be driven by enhanced transport or atmospheric mixing, which can be indicative of atmospheric dynamic processes such as the boundary-layer low-level jets (Klein et al., 2014). The enhanced nighttime ozone then reshapes the ozone diurnal cycle, and may increase daily integrated ozone exposure time to human and vegetation that threatens human health (Turner et al., 2016; Fleming et al., 2018) and crop yields (Yue et al., 2017; Lefohn et al., 2018; Feng et al., 2022; Li et al., 2022). Enhanced nocturnal ozone can also increase the oxidation capacity by stimulating the nitrate radical formation (Wang et al., 2021), and promote the formation of secondary pollutants such as particulate nitrate and secondary organic aerosols (Brown and Stutz, 2012; Huang et al., 2020). It can further affect atmospheric chemistry of the following day (Millet et al., 2016; Caputi et al., 2019; Zhao et al., 2020). While long-term trends of mean nighttime ozone level have been extensively studied as an important metric for assessing ozone air**

[revised manuscript text omitted]

**Comment [2-3]:** Similarly, I don't find the lengthy discussion of differences in frequency of occurrence of nocturnal ozone increases between the US, Europe and China compelling. In the US and Europe there are generally lower daytime $O_3$ peaks (which leaves a smaller enhancement in the residual layer) and, I think, fewer nocturnal NO emissions (which results in less complete titration at night) so that the difference between the surface and the residual layer is less stark and mixing has a smaller effect. What really matters is probably the 24 hour integrated $O_3$ exposure, which is much higher in most of China than in the EU or US, possibly exacerbated by trends of increasing NOEs over time? But not by a ton (at least not yet since the nocturnal $O_3$ is so low). And the nighttime exposures in the EU and US are comparably more important because the nighttime $O_3$ levels are higher and the daytime peaks lower (generally). You do eventually get there towards the end of the manuscript but I think the whole paper would be improved if you discuss the logical explanations for these broad differences when you are describing the observed distributions.

**Response [2-3]: Thank you for pointing it out. We attempted to organize the paper into two parts: characteristics and mechanism. The higher frequency NOE event (Figure 1a) in China than in US and Europe is a new finding that motivates us to probe into the mechanism. Then we do not start the discussion with Figures 1b-d because they are closely related to the mechanisms, instead we move to Figures 2-4 for a comprehensive description of NOE statistics (spatial pattern, seasonal difference, peak value, and evolution of other pollutant), which also provides hints on the NOE mechanisms. Sections 3.2 and 3.3 focus on the mechanism and include detailed discussion on Figures 1b-d. We believe this organization is easier for readers who are not familiar with these unusual NOE**

events. Showing Figures 1b-1c with Figure 1a allows a direct comparison between NOE frequency and ozone level, but indeed it confuses the logics. We further clarify this in the text "As shown in Figures 1b and 1c, the spatial pattern of NOE event frequencies is closely related to the afternoon (14-17 LT) ozone and nighttime $O_x$ ($O_3$+$NO_2$) concentrations measured at the surface. This feature has important implications for understanding the mechanism of NOE events, which will be analyzed in Section 3.2."

We agree with your analyses on the low NOE frequency over Europe and US. We have added them in the text "In other regions over Europe and US, we see much lower NOE event frequencies on average. This is because the daytime peak ozone is relatively lower than those over China, leading to low nighttime ozone concentrations in the residual layer (as indicated by the afternoon ozone at surface of 30 ppbv or less), in addition nighttime NO emissions there are low, which contribute to weak titration effect (as indicated by the small difference between nighttime ozone and $O_x$ level). As such, the ozone difference between the surface and the residual layer is less stark, and residual layer ozone cannot serve as an effective source to enhance ozone at the surface even there is strong mixing or transport.".

We have also discussed the influence of NOE event on integrated ozone exposure in the introduction [Response 2-2] and in Section 3.4 "While the enhanced nocturnal mixing between the residual layer and nighttime boundary layer contributes to nocturnal ozone enhancement at the surface, the enhanced ozone is also subject to more efficient chemical destruction and dry deposition, resulting in lower ozone peak values on the next day (Hu et al., 2013; Caputi et al., 2019). As such, whether NOE events would increase or decrease the ozone level and integrated ozone exposure in the following day is yet to be determined. "

Reference

Caputi, D. J., Faloona, I., Trousdell, J., Smoot, J., Falk, N., and Conley, S.: Residual layer oz one, mixing, and the nocturnal jet in California's San Joaquin Valley, Atmos. Chem. P hys., 19, 4721-4740, 10.5194/acp-19-4721-2019, 2019.

Hu, X.-M., Klein, P. M., Xue, M., Zhang, F., Doughty, D. C., Forkel, R., Joseph, E., and Fue ntes, J. D.: Impact of the vertical mixing induced by low-level jets on boundary layer ozone concentration, Atmospheric Environment, 70, 123-130, https://doi.org/10.1016/j.at mosenv.2012.12.046, 2013.

**Comment [2-4]:** Also related to point #2 above, I appreciate the usage of odd oxygen in your analysis but it seems like an afterthought right now and I think you should introduce it earlier (for example, it could logically be used in discussing the observed differences in nocturnal $O_3$ behavior between the EU, the US and China.)

**Response [2-4]: We agree. We have moved Figure S4 to Figure 1 to show the $O_x$**

**pattern. Please also kindly find our revision as stated in [Response 2-3].**

**Comment [2-5]:** NNOE (non-nocturnal ozone enhancement) is a weird acronym because it sounds like it should be an enhancement that happens during the day rather than a lack of an enhancement at night. Perhaps "non-enhanced nocturnal ozone" or "stable nocturnal ozone event" would work better?

**Response [2-5]: Thank you for pointing it out. We have adopted the "non-enhanced nocturnal ozone" as the full term for NNOE in this paper.**

**Comment [2-6]:** I would encourage the authors to think about whether certain points could be made using correlation plots rather than color-scaled maps that the reader must qualitatively compare. I had to do a lot of scrolling back and forth to see some of the trends that were being described. The top rows of Figures 1 and 2 make sense as color-scaled maps but when you are trying to compare NOE frequency to nocturnal ozone concentrations and subsequent-day afternoon $O_3$ I think those would be much better communicated by correlations. Actually the relationship between NOE frequency and afternoon ozone is less direct than looking at, for example, peak $O_3$ in a NOE compared to $O_3$ from the day before (or the following day). Why not plot those correlations instead? Similarly, I find that Figure 3 takes more effort than it should to look at. Would it communicate the same thing if you showed a single map that was colorscaled by the mean enhancement observed for evenings on which an NOE occurred? I believe the main point is that the sites that have the most frequent NOEs also experience the largest $O_3$ increases when they occur. Or perhaps that could also be a correlation plot. **Response [2-6]: Thank you for pointing it out. We have added Figure 5 (Figure R1) to quantitatively determine the relationship between NOE frequencies, afternoon ozone, and nighttime $O_x$. We now state in the text "Figures 1b and 1c present the mean afternoon ozone and nighttime $O_x$ concentrations over China, Europe, and the US. Comparison of Figure 1a and 1b/1c reveals that the spatial pattern of NOE event frequencies, afternoon ozone levels and nighttime $O_x$ level (both indicative of ozone in the nighttime residual level) are largely consistent. In particular, sites over China, Mediterranean, and mountainous western US with frequent NOE event recorded are consistently observing high afternoon ozone levels or nighttime $O_x$ level. Figure 5 further shows that NOE frequency at Chinese sites increases with rising mean afternoon ozone level or nighttime $O_x$ level, and is 10% (18%) higher when the afternoon ozone levels (nighttime $O_x$ levels) exceed 50 ppb than when they are below 30 ppb.". We also derive a significant and positive linear correlation coefficient of 0.5 (*p-value*<0.01) between NOE frequencies and nighttime $O_x$ level and 0.2 (*p-value*<0.01) between NOE frequencies and afternoon ozone level. However, we think that the value of linear correlation coefficient is somewhat misleading, because NOE is defined with a fixed threshold, so the relationship between its frequency and ozone level would not be linear. So we prefer to illustrate their relationship by comparing the mean NOE frequency at different ozone/$O_x$ range**

as shown in Figure R1. We reserve the map plot in Figure 1, because the spatial pattern itself convey additional information on where the hotspots of NOE frequency and ozone level are located.

Figure 3 aims to illustrate the frequency of different magnitudes of the NOE events at different regions in China, which may be a great concern for nighttime ozone air quality prediction. It is not for direct comparison between the NOE frequency and magnitude, so we do not prefer to replace it with a scatter plot. But we indeed find that sites that have the most frequent NOEs also experience the largest $O_3$ increases when they occur with a linear correlation coefficient of $r$=0.4. We have added the following text "We also find a significant positive correlation between NOE frequency and magnitude across 814 Chinses sites ($r$=0.4, $p$-value<0.01), indicating that sites with more frequent NOE events are more likely to experience larger nighttime ozone increase."

[Figure]

**Figure R1 (Figure 5). The relationship between NOE frequencies and afternoon ozone (a) and nighttime $O_x$ (b) at 814 Chinese sites. The colored box-and-whisker plots (5th, 25th, 50th, 75th, and 95th percentiles, and mean values denoted as triangles) show NOE frequencies at different concentrations of ozone or $O_x$.**

**Comment [2-7]:** When looking at vertical profiles (eg section 3.2 and Figure 6), I think potential temperature might show your point better in terms of highlighting the altitude range that is being affected by cooling at the surface.

**Response [2-7]: Thank you for the suggestion. We have used the profile of potential temperature to replace temperature, supporting the enhanced atmospheric vertical mixing in the NOE events. The new Figure R2 (Figure 7) is shown below. We have revised the text accordingly.**

[Figure]

**Figure R2 (Figure 7). Comparison of vertical profiles of potential temperature at five typical cities between the NOE (the top panels) and the NNOE (the bottom panels) events. The colored lines represent potential temperature profiles at different time of the night. The temperature data are from the ERA5 dataset.**

**Comment [2-8]:** I wonder if you have considered the effects of reactions between $NO_2$ and $O_3$ to form $NO_3$ and $N_2O_5$ in the surface layer? $NO_3$ is quite reactive and $N_2O_5$ has a high deposition velocity so it could be an appreciable fraction of the observed nighttime $O_3$ decreases. I would consider it part of "$NO_x$ titration of $O_3$" but I don't think I saw this process mentioned explicitly anywhere. If it has not been considered it certainly should be. $NO_3$ is highly reactive and $N_2O_5$ deposits very easily so they both could be substantial as a nocturnal $O_x$ loss.

**Response [2-8]: Yes, we have. Our original text described the effects of reactions between $NO_2$ and $O_3$ to form $NO_3$ and $N_2O_5$ in Line 206. We now also add it to Line 192: "This is reasonable because the decreasing rate of ozone is fast at early night, due to the rapid chemical loss through $NO_x$ titration ($NO+O_3 \rightarrow NO_2+O_2$, $NO_2+O_3 \rightarrow NO_3+O_2$, $NO_3+NO_2+M \rightarrow N_2O_5+M$) when ..".**

**Comment [2-9]:** Does it matter that sunrise and sunset is at a different time of day across different latitudes (or between the cold and the warm season)? It seems like defining nighttime in terms of clock time rather than solar time could bias things, especially increases that are observed in the early morning in the summer when you might have sunlight for the beginning of commuting time. I'm thinking especially of the NOE events assigned to have happened between 8 and 9 pm and 4 and 5 am.

**Response [2-9]: Thank you for pointing it out. We did not consider solar time in the analyses. The solar time for a given site may differ from the actual clock time by as much as plus or minus 45 minutes at a given time of year. Based on ozone time series shown in Figure 4, we find that ozone enhancement between 8 and 9 pm and 4 and 5 am is of relatively low frequency, suggesting that difference in solar time and clock time should not exert a large impact on our analyses. We**

**thus prefer to stick to the clock time (local time) in our analyses for easier communication to readers.**

**Comment [2-10]:** I recommend trying to cut down on the figures that accompany the case studies. I don't think the main finding, that vertical mixing can largely explain the observed NOEs is particularly controversial so I think it should be sufficient to describe briefly the particular instances that were investigated and the consistency between them but I don't think this requires the 1-2 figures per event that are currently shown.

**Response [2-10]: Thank you for the suggestion. We have moved the Figure 8 and Figure 10 to the supplementary text and revised the text accordingly.**

**Comment [2-11]:** MINOR issues: In the abstract, I was initially confused about what an annual mean frequency of 41% meant. After reading I believe that you calculate the annual frequency of NOEs for each site and then average across all sites. While I think changing to "mean annual frequency" would be slightly clearer, I would encourage the authors to also describe this number in slightly more detail to make things easier on the reader as I started wondering early on which locations were used for each dataset. I would include a reference to S1 (the map of locations) around line 92 rather than only in the paragraph before.

**Response [2-11]: Thank you for pointing it out. We have revised the text as "We find that the mean annual frequency of NOE events is 41±10% (i.e. about 140 days would experience NOE event per year) averaged over all 814 Chinese sites in 2014-2019, which is 46% larger than those over Europe and US.". We have also added a reference to Figure S1 when introducing the ozone data from Europe and the United States in Line 94.**

**Comment [2-12]:** Figure 1: I don't see how the inset shows mean and standard deviation.

**Response [2-12]: We used observations of all sites (N is the number of sites) in three regions to calculate the mean and standard deviation (mean ± standard deviation), respectively. We put them at the top of the figure. We have clarified in the figure caption "The values of regional mean and standard deviations among the N sites are shown inset at the top of each figure (mean ± standard deviation)."**

**Comment [2-13]:** Figure 2: Please label the colorscale for panels c and d. It's in ppb I think? But with the upper one % and the lower one not it is a bit confusing. Same issue with the inset as for Figure 1. In general I think insets, unless they are simply zoomed in on a particular region of the larger panel, should have their own axis labels, otherwise they are very hard to interpret.

**Response [2-13]: Thank you for pointing it out. We have revised accordingly.**

**Comment [2-14]:** Figure 4: Error bars would be good. Some of those profiles don't look super different for NOEs and NNOEs. And I would recommend that you harmonize axes for all sites in figure 4 if possible.

**Response [2-14]: Thank you for the suggestion. We have added shadings to represent standard deviation of hourly ozone. We do not apply the shadings to $NO_2$ and CO because it would be too crowded. We prefer to have different scales for axis because it allows a clearer examination of the evolution of air pollutants, while comparison across sites is not the focus of this plot.**

**Comment [2-15]:** Line 228, I find this sentence confusing. I can see how surface layer $O_x$ should be comparable to residual layer $O_x$. And surface layer $O_x$ would be similar to residual layer $O_3$ if $NO_2$ were a small fraction of the residual layer $O_x$ but do we know that a priori? Also, I don't think nighttime emissions of NO need be small for this relationship to hold because it simply converts $O_3$ into $NO_2$ on a one to one basis and thus conserves $O_x$.

**Response [2-15]: Thank you for correction. We have revised this sentence accordingly in Line 244 as "Similarly we may use the surface nighttime $O_x$ ($O_3$+$NO_2$) concentration as an indicator of ozone in the nighttime residual layer (Kleinman et al., 2002; Wang et al., 2018; Tan et al., 2021), assuming that nighttime titration is much stronger than the effect of deposition and regional transport."**

[Figure]

**Figure R3. The absolute values of nighttime friction velocity (U\*) and planetary boundary layer height (PBLH) averaged over all NOE and NNOE events in Beijing and Shanghai. Error bars represent the standard error of U\* and PBLH at different time.**

**Comment [2-17]:** Figure 6 – I believe this must be model data given the smoothness of the lines and your previous use of U* and PBLH from the model. But I think it would be worth clarifying that here.
**Response [2-17]: Thank you for pointing it out. The temperature, U\* and PBLH data are all from the ERA5 dataset as stated in Section 2.2. We have added this information in the figure caption for clarification.**

**Comment [2-18]:** Your text goes right from Figure 9 to Figure 11. I recommend moving Figure 10 to wherever it is that you discuss it or removing it if it is not currently discussed in the text.
**Response [2-18]: We have moved Figure 10 to the supplementary text following your suggestion.**

**Comment [2-19]:** Figure S1: the red dots are nearly invisible. Recommend marking with stars or some other symbol that will stand out in both shape and color and making them a bit bigger.
**Response [2-19]: We have enlarged the dots following your suggestion.**

**Comment [2-20]:** Figure S2: The legend says that the inset shows the number of sites with positive trend but I don't really understand what I'm looking at. As displayed I don't think these are useful and, since I don't know what you're trying to communicate, I can't figure out how to help.

**Response [2-20]: Thank you for pointing it out. We agree that this figure does not add convincing conclusion. We have removed this figure to avoid confusion.**

**Comment [2-21]:** English language – quite a few instances including from the first few pages (but not limited to):

line 62, threat should be threaten

top of p3: only one or A few and A comprehensive view on (del "the") general characteristics and mechanisms of (del "the")…

Line 71: six years OF ozone…

**Response [2-21]: Thank you for pointing it out. We corrected them accordingly.**

---

## Author Response (AR2)

Dear Editor Dr. Graciela Raga,
Thank you very much for handling our manuscript. We have addressed all the comments raised by both reviewers and incorporated them in the latest manuscript.

Sincerely,
Cheng He et al.
* * *
**Reviewer #1**

**Comment [1-1]:** It is nice to see the probability distributions of nocturnal delta[$O_3$]/delta[t] (Figure S2). But I think you counted all nighttime delta[$O_3$]/delta[t] values so that only 7.7% of them are >5 ppb/hour. Since these 7.7% of cases are mostly out of range of two-sigma standard deviation, you think they are not likely normal fluctuations. This is better than an arbitrary threshold but still is not based on the spectrum structure, which may be driven by atmospheric processes. I am not sure if you can obtain a useful spectrum structure by only including the maximum delta[$O_3$]/delta[t] value from each night, but I think it is worth trying. If you get nothing more useful, a threshold based on Figure S2 is acceptable and at least more convincing than just following the previous studies.

**Response [1-1]: Thank you for the reply and suggestion. We have further derived the probability distributions of daily maximum nocturnal delta[O3]/delta[t]. Results are shown in Figure R1. We find that 93% of the maximum delta[O3]/delta[t] values are positive, with a peak probability in the range of 2-3 ppbv hour$^{-1}$, 5 ppbv hour$^{-1}$ places as the approximate turning point that halves the probability distribution, further supporting that the 5 ppbv hour$^{-1}$ would be an appropriate threshold.**

**We further rephrase in Section 2.3: "Following previous studies of Eliasson et al. (2003) and Zhu et al. (2020), we define a nocturnal ozone enhancement (NOE) event if ozone concentration at a site increases by more than 5 ppbv ($\Delta O_3/\Delta t > 5\ ppb\ hour^{-1}$) in one of any two adjacent hours in the nighttime period. We find that nocturnal $\Delta O_3/\Delta t$ values at Chinese sites generally follow the Gaussian distribution. For all hourly nocturnal $\Delta O_3/\Delta t$ values, $\Delta O_3/\Delta t > 5\ ppb\ hour^{-1}$ cases only account for 7.7% of the $\Delta O_3/\Delta t$ dataset, indicating that this threshold should have effectively ruled out nocturnal ozone fluctuations occur under normal atmospheric conditions (Figure S2a). For the daily maximum nocturnal $\Delta O_3/\Delta t$ values only, the probability peaks in the range of 2-3 ppbv hour$^{-1}$, and the 5 ppbv hour$^{-1}$ threshold places as the approximate turning point that halves the probability distribution (Figure S2b). We thus apply the $\Delta O_3/\Delta t > 5\ ppb\ hour^{-1}$ threshold in the NOE definition."**

[Figure]

**Figure R1 Probability density distribution of daily maximum nocturnal ozone fluctuation ($\Delta O_3/\Delta t$) at all Chinese sites (represented by each curve) in 2014-2019.**

**Comment [1-2]:** Another thing I find interesting is that there are small but not negligible probabilities with substantial negative delta[$O_3$]/delta[t] values. Normally, nighttime chemistry and dry deposition cause gradual decrease of $O_3$ concentrations. However, very steep dropping of $O_3$ concentrations may occur only under some unusual atmospheric conditions. But this is out of the scope of this paper.

**Response [1-2]: Thank you for pointing it out. Nighttime titration is strong in China because of high $NO_x$ emission. The deep decrease of nighttime ozone concentrations may be related to rapid chemical loss through $NO_x$ titration. Of course, it may also be related to some unusual atmospheric conditions that effectively scavenge ozone the as the reviewer suggested. We agree that this interesting scientific question is worth studying further in the future.**
* * *
**Reviewer #2**

**Comment [2-1]: Major comments:** The paper presents an overview of the general characteristics of nocturnal ozone enhance (NOE) events in China and tries to find possible mechanisms leading to the NOE events based on 6-year observational data from the national monitoring network of China. In general, this is an excellent and well written paper. Based on statistics on friction velocity, boundary layer height and low-level jet, the correlation between the boundary layer physical processes and the NOE events is well explained, which is of great significance for improving the scientific understanding of NOE. The reviewer noted that the author has made appropriate

revisions to the paper according to the comments from other reviewers, which significantly improved the quality and publish ability of the paper. Based on the above points, the reviewer suggests accepting the paper after some minor revisions:

**Response [2-1]: We thank the reviewer for the positive and valuable comments. All of revisions have been implemented in the latest manuscript.**

**Comment [2-2]:** In the two paragraphs where Lines 280-290 are located, the explanation on the mechanism of friction velocity and PBLH causing NOE could be further enhanced: (1) The physical meaning and formula of friction velocity, and its relationship with turbulence intensity and vertical mixing. (2) The method used to calculate the PBLH (how it is derived from ERA5 model data?), the relationship between the increase of PBLH and the downward infiltration of residual layer mass.

**Response [2-2]: Thank you for pointing it out. We have modified the text to describe the meaning friction velocity and PBLH and their implication for the mechanisms. We do not calculate the friction velocity and PBLH as they are available from the ERA5 dataset.**

**We state sources of the data in Section 2.2: "We apply three-dimensional fields of meteorological parameters including temperature, relative humidity, horizontal and vertical wind speed and direction on pressure levels, and two-dimensional fields of planetary boundary layer height, and friction velocity from the ERA5 dataset, i.e. the fifth generation of the European Centre for Medium-Range Weather Forecasts (ECMWF) atmospheric reanalysis of the global climate (https://cds.climate.copernicus.eu/#!/home, last access: 15 April 2022)."**

**We add the following statement to illustrate the relationship with increasing vertical mixing and the U\* / PBLH in Line 280: "The U\* and PBLH are applied to assess the atmospheric turbulence capacity for the vertical mixing, transport and diffusion of air pollutants. Increasing U\* and PBLH levels typically indicate enhanced turbulent kinetic energy and intensity, thus a more unstable boundary layer and stronger atmospheric mixing (Ren et al., 2021; He et al., 2022).", and in Line 289: "More importantly, we see distinct differences in their temporal evolution. U\* and PBLH typically show a steady decreasing trend throughout the nighttime during NNOE events, while the U\* and PBLH show increase in at least a certain part of the nighttime period in the NOE events, suggesting that atmospheric mixing and downward infiltration of residual layer mass are becoming more active (Figure 6a and 6c)."**

Reference

Ren, Y., Zhang, H., Zhang, X., Wei, W., Li, Q., Wu, B., Cai, X., Song, Y., Kang, L., and Zhu, T.: Turbulence barrier effect during heavy haze polluti on events, Sci Total Environ, 753, 142286, 10.1016/j.scitotenv.2020.142286, 2021.

He, J. Y., Chan, P. W., Li, Q. S., Li, L., Zhang, L., and Yang, H. L.: Observ ations of wind and turbulence structures of Super Typhoons Hato and Ma

ngkhut over land from a 356 m high meteorological tower, Atmospheric Research, 265, 10.1016/j.atmosres.2021.105910, 2022.

**Comment [2-3]:** It is suggested to further enhance the literature review of NOE related issues, although the current one is excellent enough. In the past few months, some new studies on the characteristics and mechanisms of nocturnal ozone pollution have been published, and some of the conclusions of these new references can also support the conclusions of this paper.

**Response [2-3]: Thank you for pointing it out. We notice that two recently published papers that systematically studied the characteristics of nocturnal ozone enhancement, and quantified the impact of vertical and horizontal transport in the PRD region. We have added them in Line 316-367.**

**Reference**

Wu, Y., Chen, W., You, Y., Xie, Q., Jia, S., and Wang, X.: Quantitative impacts of vertical transport on long-term trend of nocturnal ozone increase over the Pearl River Delta region during 2006-2019, Atmos. Chem. Phys. Discuss., 2022, 1-29, 10.5194/acp-2022-360, 2022.

Yang, H., Lu, C., Hu, Y., Chan, P.-W., Li, L., and Zhang, L.: Effects of Horizontal Transport and Vertical Mixing on Nocturnal Ozone Pollution in the Pearl River Delta, Atmosphere, 13, 10.3390/atmos13081318, 2022.

**Comment [2-4]:** Please verify or correct some minor flaws in the text editing. For example, it is recommended to change "six-year observation" to "6-year observation". Remove the underline from the text near L400…

**Response [2-4]: Thank you for pointing it out. We corrected them accordingly.**
* * *
**Comments from Editor Dr. Polina Shvedko:** For the next revision, please check if your figures containing maps/aerial images require a copyright statement/image credit and add it to the figures (or captions) (https://publications.copernicus.org/for_authors/manuscript_preparation.html#mapsaerials). If these figures were entirely created by the authors, there is no need to add a copyright statement or credit. In that case it is important that you confirm this explicitly by email. 2. Please ensure that the colour schemes used in your maps and charts allow readers with colour vision deficiencies to correctly interpret your findings. Please check your figures using the Coblis – Color Blindness Simulator (https://www.color-blindness.com/coblis-color-blindness-simulator/) and revise the colour schemes accordingly.

**Response: Thank you very much for the suggestion. We have revised accordingly.**